# Interlayer-Expanded MoS_2_ Enabled by Sandwiched Monolayer Carbon for High Performance Potassium Storage

**DOI:** 10.3390/molecules28062608

**Published:** 2023-03-13

**Authors:** Yuting Zhang, Lin Zhu, Hongqiang Xu, Qian Wu, Haojie Duan, Boshi Chen, Haiyong He

**Affiliations:** 1Ningbo Institute of Materials Technology and Engineering, Chinese Academy of Sciences, Ningbo 315201, China; 2University of Chinese Academy of Sciences, Beijing 101400, China

**Keywords:** potassium-ion batteries, anode, layered 2D materials, transition-metal dichalcogenides, MoS_2_

## Abstract

Potassium-ion batteries (PIBs) have aroused a large amount of interest recently due to the plentiful potassium resource, which may show cost benefits over lithium-ion batteries (LIBs). However, the huge volume expansion induced by the intercalation of large-sized potassium ions and the intrinsic sluggish kinetics of the anode hamper the application of PIBs. Herein, by rational design, nano-roses assembled from petals with a MoS_2_/monolayer carbon (C-MoS_2_) sandwiched structure were successfully synthesized. The interlayer distance of ultrathin C-MoS_2_ was expanded from original MoS_2_ of 6.2 to 9.6 Å due to the formation of the MoS_2_-carbon inter overlapped superstructure. This unique structure efficiently alleviates the mechanical strain, prevents the aggregation of MoS_2_, creates more active sites, facilitates electron transport, and enhances the specific capacity and K^+^ diffusion kinetics. As a result, the prepared C-MoS_2_-1 anode delivers a high reversible specific capacity (437 mAh g^−1^ at 0.1 A g^−1^) and satisfying rate performance (123 mAh g^−1^ at 6.4 A g^−1^). Therefore, this work provides new insights into the design of high-performance anode materials of PIBs.

## 1. Introduction

The development of post lithium-ion batteries (LIBs) is important to deal with the increasing energy demands because of the scarcity of lithium resources. Potassium-ion batteries (PIBs) and sodium-ion batteries (SIBs) have recently aroused a large amount of interest in large-scale energy storage applications due to the abundance of potassium and sodium, which is expected to be more cost effective [1]. Compared with Na/Na^+^ (−2.73 V vs. standard hydrogen electrode (SHE)), the potential of K/K^+^ (−2.93 V vs. SHE) is close to that of Li/Li^+^ (−3.04 V vs. SHE), resulting in the increase in the operating voltage, which provides an opportunity to achieve the high energy density of PIBs. The smaller Stokes radius of K^+^ (3.6 Å) results from the weaker Lewis acidity than in the cases of Li^+^ (4.8 Å) and Na^+^ (4.6 Å) enables K^+^ to have a higher transport mobility in the electrolyte, which indicates that PIBs possess the potential for higher power density [2]. In addition, although it is possible potassium dendrites to occur that are similar to lithium dendrites in PIBs under the overcharged state, the lower melting point of potassium metal (63.4 °C) than that of lithium metal (180.5 °C) will lead to dendritic potassium, which would melt readily before creating a short-circuit, leading to a much better safety capability of PIBs. Furthermore, aluminum (Al) and potassium do not form alloy compounds, thus, Al foil can be used as the current collector in PIBs to replace copper (Cu) foil, which reduces both the weight as well as the economic cost of the electrode [3]. Therefore, these merits make PIBs one of the most promising alternatives to LIBs, particularly for large-scale energy storage applications.

Up to now, a variety of anode materials for PIBs have drawn special attention including carbonaceous materials (graphite, soft carbon, hard carbon, graphenes, carbon nanotubes) [4,5,6,7], alloy types materials (bismuth, stannum) [8,9], and conversion type materials (chalcogenide compounds) [10,11]. Up until now, graphite-based anodes have been extensively reported because of the low-costs, safety, and low voltage plateau for PIBs [12]. In spite of the above advantages, the theoretical capacity of graphite is only 279 mAh g^−1^, and the sluggish diffusion kinetics of K^+^ in graphite hinder the rate capability for PIBs [13]. Consideration of the fast potassiation/depotassiation kinetics and relatively high charge/discharge capacities of transition metal dichalcogenides (TMDs), TMDs have become a class of promising alternatives [14]. As a typical TMD, MoS_2_ has a two-dimensional (2D) layered structure in which sulfur atoms are interconnected with molybdenum atoms to form hexagons within the layers and weak van der Waals forces act between the layers. Due to the anisotropic structure for MoS_2_, permeable channels and substantial amounts of reaction sites facilitate the insertion/extraction of potassium ions. Ren et al. first studied bare MoS_2_ as a host material for K^+^ electrochemical intercalation, and the MoS_2_ delivered a reversible capacity of 65 mAh g^−1^ at 20 mA g^−1^ after 200 cycles [15]. The inferior cycle stability of MoS_2_ for PIBs is due to limited interlayer distance between the adjacent MoS_2_ monolayer and MoS_2_ tends to aggregate when undergoing volume change during charge/discharge cycles. Moreover, the intrinsically low electronic conductivity of MoS_2_ leads an to unsatisfactory rate performance because of the large polarization [16]. Thereby, new approaches and structures are necessary when seeking to further the improvement of MoS_2_ electrodes.

Recently, strategies of enlarging the interlayer distance of adjacent MoS_2_ monolayers, fabricating few-layer MoS_2_, and decorating MoS_2_ with carbon materials have been reported to improve the K^+^ storage performance. For example, Wang et al. reported few-layered 2H MoS_2_ nanosheets with an expansion of the interlayer spacing by liquid-phase exfoliation in water as the PIB anode, which delivered a high capacity of 203 mAh g^−1^ at 200 mA g^−1^ after 300 cycles and a low capacity decay of 0.02% per cycle over 1500 cycles [17]. Hu et al. synthesized a yolk-shell structured hollow porous carbon-sphere-confined MoS_2_ composite (MoS_2_@HPCS) as a PIB anode. The high reversible capacity of 254 mAh g^−1^ was obtained at 500 mA g^−1^ after 100 cycles and maintained 126 mAh g^−1^ over 500 cycles at a current density of 1000 mA g^−1^ [18].

However, the contact interface between MoS_2_ and carbon is limited in several reports. Interestingly, Jia and Cui et al. respectively reported the MoS_2_-based structure with expanded interplanar spacing by the intercalation of conductive carbon through in situ carbonation of organic compounds (oleylamine (OAm), ethylene glycol (EG), and dopamine) [16,19]. The carbon layer in direct contact with the MoS_2_ sheets results in maximizing the interface between the MoS_2_ layers and carbon layer, which has been considered beneficial for ion and electron transport [20]. Inspired by the strategies, positively charged poly diallyl dimethyl ammonium chloride (PDDA) has been reported to decorate metal–organic frameworks (MOFs) and reduced graphene oxide (rGO) to gain a positive charge surface [21]. A negatively charged molybdenum source can easily be adsorbed onto the PDDA. After hydrothermal and annealing treatment, the monolayer carbon derived from PDDA can intercalate between MoS_2_ layers to expand the interplanar spacing of MoS_2_.

Herein, we demonstrate a novel approach to fabricate nano-roses assembled from a MoS_2_ and carbon monolayer (C-MoS_2_) sandwiched structure nano-petals by PDDA-assisted hydrothermal and annealing treatment. The (002) interlayer spacing of C-MoS_2_ nano-petals was significantly expanded as large as 9.6 Å. Compared with the bare MoS_2_, the C-MoS_2_ helped to conquer the shortcomings in the following aspects. First, the enlarged interlayer distance of ultra-thin MoS_2_ is in favor of alleviating the mechanical stress during the charge/discharge cycles to maintain the structural integrity. Second, direct contact between MoS_2_ and the single-layer carbon sheet, on one hand, is beneficial for electron transport to allow for faster potassiation/depotassiation kinetics, and on the other hand, effectively prevents the aggregation of MoS_2_ to improve the cycling stability. Third, the few-layer MoS_2_ with expended distance and carbon derived from PDDA creates more active sites to improve the capacity and rate performance of C-MoS_2_. As a result, the C-MoS_2_ sandwiched structure well-addressed the poor cycling stability and low rate performance of MoS_2_ for PIBs. The C-MoS_2_-1 electrode exhibited high rate performance (123 mAh g^−1^ at 6.4 A g^−1^), which was 11 times higher than that of bare MoS_2_ (11 mAh g^−1^). C-MoS_2_-1 delivered a specific capacity of 437 mAh g^−1^ at 0.1 A g^−1^, which was 417 mAh g^−1^ higher than MoS_2_. At the high current density of 1.0 A g^−1^, the capacity of C-MoS_2_-1 was maintained over 273 mAh g^−1^ after 100 cycles, which was more than 10 times higher than that of bare MoS_2_ (25 mAh g^−1^).

## 2. Results and Discussion

The synthetic strategy of C-MoS_2_-1 involves the hydrothermal reaction and annealing treatment. First, poly dimethyl diallyl ammonium chloride (PDDA), as a conductive cationic polymer, can be strongly adsorbed on Mo_7_O_24_^6−^ anions by electrostatic attraction. In the subsequent hydrothermal process, Mo_7_O_24_^6−^ reacts with thiourea to generate MoS_2_. After the heat treatment, a C-MoS_2_ hybrid nanosheet is obtained from the crystallization of MoS_2_ and carbon degradation from PDDA, intercalating into MoS_2_ layers. The hybrid samples were named C-MoS_2_-X, and X is the gram weight of the PDDA aqueous solution added during sample preparation.

The crystallographic characteristics of the MoS_2_ and C-MoS_2_ samples were characterized by X-ray diffraction (XRD). As shown in Figure 1a, the bare MoS_2_ material showed diffraction peaks at 14.3°, 33.3°, and 59.1°, corresponding to the (002), (100), and (110) crystalline planes of 2H MoS_2_ (JCPDS#37-1492) [22]. According to Bragg’s formula (nλ = 2dsin θ), the distance of the (002) lattice plane can be calculated to be about 6.2 Å. Interestingly, as the content of PDDA increased, the sharp peak located at 14.3° gradually decreased and a dispersive peak at 9.2° gradually appeared, which indicates that the distance between the (002) crystal plane of MoS_2_ can be enlarged from about 6.2 Å to 9.6 Å by the intercalation of carbon [23,24]. Moreover, a small peak located at 18.0° gradually appeared as the content of PDDA increased, and the interplanar distance was calculated to be about 4.8 Å according to Bragg’s formula. This distance is approximately half of the distance of expended (002) lattice plane, which indicates the distance between MoS_2_ and the graphene layer (Figure 1b). The approximate diploid relation verifies the MoS_2_ and graphene monolayer inter overlapped stacked superstructure [25]. Interlayer expansion of C-MoS_2_ is very attractive due to improved reaction kinetics, alleviating the strain and improving the cycling stability during the charge/discharge process. Aside from peak shifting, the diffraction peaks of expanded (002) were much broader than that of bare MoS_2_, which could be attributed to the presence of carbon, inhibiting the growth of crystalline MoS_2_ [16].

Raman spectra was used to further characterize the composition and crystallinity information of the MoS_2_ and C-MoS_2_ samples. As shown in Figure 1c, two characteristic peaks located at around 384 and 413 cm^−1^ of bare MoS_2_ were attributed to the Mo-S in-plane $E_{2g}^{1}$ and out-of-plane A_1g_ vibrational modes [26]. As for C-MoS_2_, the A_1g_ blue shifted to 401 cm^−1^, which illustrates the thickness of MoS_2_ decreased after adding PDDA. Compared to bare MoS_2_, the inter-peak separation of C-MoS_2_ between $E_{2g}^{1}$ and A_1g_ significantly decreased from 29 to 24 cm^−1^, further confirming the formation of few-layer MoS_2_ [27]. Few-layer MoS_2_ is advantageous for accelerating K^+^ diffusion and decreasing mechanical strain during K^+^ intercalation/deintercalation. Moreover, the characteristic D-band (sp^3^-hybridized amorphous carbon) and G-band (sp^2^-hybridized graphitic carbon) were observed at approximately 1355 and 1582 cm^−1^ from C-MoS_2_-0.5 and C-MoS_2_-1, respectively. The ratio of the relative intensities of the D- and G-bands (I_D_/I_G_) reflects the defect concentration of carbon materials [28]. The calculated I_D_/I_G_ value of C-MoS_2_-1 (I_D_/I_G_ = 1.295) was higher than that of C-MoS_2_-0.5 (I_D_/I_G_ = 1.198), indicating a higher defect concentration of the former. These defects mainly result from PDDA derived amorphous carbon, which has been reported to improve electron conductivity [29].

The morphology and microstructure of C-MoS_2_-1 were investigated by scanning electron microscopy (SEM) and transmission electron microscopy (TEM) analyses (Figure 2). As shown in Figure 2a,b, it can be seen that flower-like MoS_2_ consists of nanosheets, and the diameter of flower-like structures is usually in the range of 400 to 600 nm. As presented in Figure 2c,d, the TEM images for C-MoS_2_-1 indicate that the interlayer spacing of MoS_2_ was about 10.0 Å, and the layer number of MoS_2_ nanosheets was almost in the range of 2 to 6, which is in accordance with the XRD and Raman observations. Therefore, the intercalation of monolayer carbon via organic PDDA molecule degradation would be a new strategy to fabricate few-layer C-MoS_2_ with expanded interlayer spacing. The expanded interlayer distance would boost the reaction kinetics, and few-layer C-MoS_2_ is in favor of alleviating the mechanical stress to maintain the structural integrity. As shown in Figure 2e–h, high-angle annular dark-field (HAADF) microscopy equipped with energy dispersive spectroscopy (EDS) revealed the homogeneous distribution of C, Mo, and S in C-MoS_2_-1.

The chemical states of C-MoS_2_-1 were tested by X-ray photoelectron spectroscopy (XPS) (Figure 3a–d). In the high-resolution C 1s spectra (Figure 3a), three peaks centering at 284.8, 286.4, and 288.4 eV correspond to C–C, C–O, and C–N bond, respectively [17]. In Figure 3b, the high-resolution S 2p spectra were deconvoluted into two separate peaks of S 2p_1/2_ and S 2p_3/2_ at 163.3 and 162.1 eV, respectively. In the case of the Mo 3d spectrum in Figure 3c, two strong peaks at 229.3 and 232.5 eV as well as two weak peaks at 226.5 and 235.6 eV belonged to S-Mo 3d_5/2_, S-Mo 3d_3/2_, S 2s, and Mo 3d_5/2_. The S 2p and Mo 3d spectra exhibit typical 2H-MoS_2_ features [30]. Figure 3d revealed the strong peak at 395.2 eV ascribed to Mo 3p_3/2_, and the existence of the pyridinic N and pyrrolic N at 398.0 and 400.2 eV, respectively, which has been reported to be beneficial for ion and electron transport [31]. The Brunauer–Emmett–Teller (BET) surface area of the C-MoS_2_-1 material was calculated to be 25.64 m^2^ g^−1^ (Figure 3e). The large specific surface area facilitates the contact of the electrolyte with a more active material, thus shortening the ion diffusion distance, enhancing K^+^ insertion/extraction within individual MoS_2_ monolayers, and accommodating volume change [32]. Moreover, C-MoS_2_-1 shows an IV isotherm with a hysteresis loop, which indicates the existence of mesopores in C-MoS_2_-1. Meanwhile, the pore size distribution of C-MoS_2_-1 mostly lies in 2 to 50 nm according to the Barrett–Joyner–Halenda (BJH) pore-size distribution curve (Figure 3f). The diffusion length for potassium ions would be shortened due to the mesoporous structure.

Cyclic voltammetry (CV) curves were recorded to precisely analyze the charge and discharge properties of the C-MoS_2_-1 electrode in PIBs (Figure 4a). The peak at 1.2 V was related to K^+^ intercalation into MoS_2_ layers to form K_x_MoS_2_ [33]. The weak and broad peak that appeared at about 0.8 V may be ascribed to the formation of SEI. When the potential decreased below 0.5 V, K_x_MoS_2_ subsequently underwent a conversion reaction from K_x_MoS_2_ to Mo and K_2_S, which is in contrast with the peak near 1.7 V for the formation of MoS_2_ in the anodic scan [34]. The CV curves of the C-MoS_2_-1 electrode overlapped each other except for the first cycle, indicating great reversibility. The discharge and charge processes of the C-MoS_2_-1 electrode in PIBs can be expressed as follows:MoS_2_ + xK^+^ + xe^−^ = K_x_MoS_2_
K_x_MoS_2_ + (4 − x) K^+^ + (4 − x) e^−^ = Mo + 2K_2_S

Figure 4b shows the galvanostatic charge–discharge (GCD) profiles of C-MoS_2_-1 at a current density of 0.1 A g^−1^. The initial coulombic efficiency (ICE) of C-MoS_2_-1 was 57.7%, which was 0.5% higher than that of bare MoS_2_. The capacity loss at the first cycle was mainly due to the irreversible electrolyte reduction by the formation of the solid-electrolyte interphase (SEI) and the partially reversible redox reaction of MoS_2_ in the first discharge/charge cycle.

Electrochemical performances of C-MoS_2_ are exhibited compared to the bare MoS_2_ for PIBs in Figure 4c–g. The C-MoS_2_-1 electrode delivers a high reversible capacity of 413 mAh g^−1^ after 50 cycles at a current density of 0.1 A g^−1^. In contrast, the discharging capacities of the MoS_2_ and C-MoS_2_-0.5 electrodes were only 6 and 34 mAh g^−1^, respectively (Figure 4c). The increased capacity of C-MoS_2_-1 is mainly due to the increased spacing of the few-layer MoS_2_ and the addition of carbon, providing more reactive sites. As displayed in Figure 4d, C-MoS_2_-1 exhibited the best rate capability. At the stepwise augmenting current densities from 0.1, 0.2, 0.4, 0.8, 1.6, 3.2 to 6.4 A g^−1^, C-MoS_2_-1 delivered high specific capacities of 384, 368, 349, 312, 230, 167, and 123 mAh g^−1^. Nevertheless, the specific capacity of MoS_2_ decayed severely from 281 to 15 mAh g^−1^ under the same testing condition. When the current density switched back to 0.1 A g^−1^, the capacity of C-MoS_2_-1 remained at 437 mAh g^−1^, which was 417 mAh g^−1^ higher than that of the bare MoS_2_. The superior potassium storage properties of C-MoS_2_-1 not only result from the enlarged interlayer distance of few-layer MoS_2_, but also the improvement in the conductance. The variation in the polarization voltage of MoS_2_ and C-MoS_2_ at various current densities is an important manifestation of dynamic excellence. As shown in Figure 4e, at the stepwise current densities from 0.1, 0.2, 0.4, 0.8, 1.6, 3.2 to 6.4 A g^−1^, the polarization voltage of C-MoS_2_-1 was always smaller than that of MoS_2_ and C-MoS_2_-0.5. Moreover, the enhancive polarization voltage variation of C-MoS_2_-1 between 0.1 and 6.4 A g^−1^ was 0.83 V. In comparison, the polarization voltage increment of MoS_2_ and C-MoS_2_-0.5 was between 5.66 V and 4.33 V in the same case, which was 6.8 and 5.2 times as much as that of C-MoS_2_-1, respectively. The apparent superiority of C-MoS_2_-1 in the reaction kinetics may benefit from a unique MoS_2_-carbon inter overlapped superstructure. This structure efficiently facilitates electron transport and enhances the K^+^ diffusion kinetic, which greatly enhances the K^+^ transport rate in C-MoS_2_-1. In comparison with the other reported MoS_2_-based anodes for PIBs, C-MoS_2_-1 presented an outstanding rate performance (Figure 4f) [16,17,18,35,36,37,38,39]. At a current density of 1.0 A g^−1^, the superiority of C-MoS_2_-1 was more satisfactory (Figure 4g). The C-MoS_2_-1 electrode showed a reversible capacity of 273 mAh g^−1^ after 100 cycles, which was more than ten times higher than that of MoS_2_ and C-MoS_2_-0.5. It is worth mentioning that the specific capacity showed an increase followed by a decrease of C-MoS_2_-1 both at 0.1 A g^−1^ and 1.0 A g^−1^, which is speculated to be the gradually expanded interlayers of MoS_2_ during charge and discharge, resulting in enriched active sites for K^+^ storage, thus specific capacity increased at the beginning.

Electrochemical impedance spectroscopy (EIS) recorded a frequency range from 10^5^ Hz to 10^−2^ Hz (Figure 5a). In the high-medium frequency region, it can be seen that the C-MoS_2_-1 electrode delivered the smallest diameter of the semicircle, representing decreased charge-transfer resistance because of the presence of carbon. In the low frequency region, the C-MoS_2_-1 electrode also presents a high slope angle than MoS_2_ and C-MoS_2_-0.5, suggesting a better K^+^ diffusion in the bulk electrode of C-MoS_2_-1 [40].

The galvanostatic intermittent titration technique (GITT) was used to further evaluate the diffusion kinetics at different potassiation and depotassiation states in the C-MoS_2_-1 electrodes (Figure 5b). The K^+^ coefficient (D) can be obtained according to the equation [41]:$$D=\frac{4}{\pi \tau }{\left(\frac{V_{B}m_{B}}{{\mathrm{SM}}_{B}}\right)}^{2}{\left(\frac{\Delta E_{s}}{\Delta E_{t}}\right)}^{2}$$
where *τ* is the relaxation time; *S*, m*_B_*, M*_B_*, and *V_B_* denote the geometric area of the electrode, mass, molar mass, and molar volume of the electrode material; Δ*E_t_* and Δ*E_S_* represent the voltage change during the current pulse and steady-state process.

According to the GITT analysis, the D of the C-MoS_2_-1 electrode was calculated to almost range from 10^−10^ to 10^−11^ cm^2^ s^−1^ during the potassiation and depotassiation process (Figure 5c,d). In the discharge process, the D of K^+^ in the initial discharge was 1.2 × 10^−10^ cm^2^ s^−1^ and then increased gradually. This phenomenon was attributed to the fast insertion of K^+^ into the expanded interlayer MoS_2_ to form K_x_MoS_2_. When the discharge potential dropped to 0.7 V, the conversion reaction began and the D of K^+^ decreased gradually. Notably, the D rapidly decreased to 5.5 × 10^−11^ cm^2^ s^−1^ at the end of the conversion reaction, with a voltage-dependent feature at this stage [42]. Figure 5d shows the D of K^+^ in the charging process, where the D of K^+^ in the initial charge was 5.3 × 10^−5^ cm^2^ s^−1^ and then gradually decreased because of the slow conversion reaction of K_2_S with Mo. The D of K^+^ increased gradually when the charge potential lifted to about 1.2 V, which may also relate to the deintercalation of K^+^ from K_x_MoS_2_.

## 3. Materials and Methods

Materials synthesis: All chemical reagents used were without further purification in this work. First, 0, 0.5, 0.6, 0.8, and 1.0 g of a PDDA aqueous solution (20 wt%, Mw: 100,000–200,000, Aladdin, Beijing, China) were dispersed in 20 mL of deionized water. Then, 400 mg of (NH_4_)_6_Mo_7_O_24_·4H_2_O powder (AR, Hushi, Shanghai, China) was dissolved in 10 mL of deionized water followed by the dropwise addition of the PDDA aqueous solution to obtain a milk-white homogeneous suspension. After 30 min, 1.0 g thiourea (99%, Zhanyun, Shanghai, China) was dissolved in the obtained suspension. The synthesis time was the same for all samples. The as-prepared suspension was transferred into a Teflon-lined autoclave and maintained at 200 °C for 20 h. The black suspension was washed to neutral and freeze-dried to obtain the precursor. Finally, C-MoS_2_ was obtained by heat-treating the precursor at 800 °C for 2 h with a ramping rate of 3 °C min^−1^ in an argon flow (100 sccm) tube furnace. The samples were named MoS_2_, C-MoS_2_-0.5, and C-MoS_2_-1 according to the gram weight of the PDDA aqueous solution added during the preparation.

Material characterization: XRD patterns were taken using an X-ray diffractometer (D8 Advance DaVinci, Bruker, Bremen, Germany) with Cu Kα radiation (λ = 1.5406 Å). Raman spectra were collected on a micro-Raman spectrometer (inVia-reflex, Renishaw, Kingswood, UK) using a laser wavelength (532 nm) with a 100% filter. SEM images and TEM images were conducted using a scanning electron microscope (S4800, HITACHI, Tokyo, Japan) and transmission electron microscope (Talos F200x, Thermo Fisher, Waltham, MA, USA). XPS analysis was implemented by a photoelectron spectrometer (AXIS ULTRA DLD, Shimadzu, Kyoto, Japan). The specific surface area and pore size distribution of the sample were evaluated by an N_2_ adsorption/desorption analyzer (ASAP-2020M, Micrometric, GA, USA).

Electrochemical performance measurements: The slurry was prepared by mixing 80 wt% active material, 10 wt% carbon black, and 10 wt% polyvinylidene in N-methyl-2-pyrrolidone (NMP). Then, the blade was coated with a copper foil and dried at 120 °C in a vacuum oven for 12 h. The working electrodes were further fabricated by cutting into disks with a diameter of 12 mm. The electrolyte was 0.8 mol L^−1^ KPF_6_ in ethylene carbonate and diethyl carbonate (EC: DEC = 1:1 in volume). Pure potassium metal pieces were used as the counter electrode and the glass fiber mat was used as the separator. With these components, CR 2016 coin-type cells were assembled in an argon-filled glove box.

The CV and EIS tests were carried out on a 1470 E electrochemical workstation (Solartron Metrology, Bognor Regis, UK). EIS was measured over the frequency range from 10^5^ Hz to 10^−2^ Hz with a sinusoidal voltage amplitude of 10 mV. The GCD and GITT tests were performed on a CT2001A measurement system (LAND, Wuhan, China) in the voltage range of 0.01–3.00 V (vs. K/K^+^) at room temperature.

## 4. Conclusions

In summary, an interesting induced growth of carbon on the interlayer of MoS_2_ (C-MoS_2_) was successfully achieved under the assistance of PDDA by the hydrothermal reaction and annealing treatment. The interlayer distance of few-layer C-MoS_2_ was expanded from 6.2 to 9.6 Å due to the formation of the MoS_2_-carbon inter overlapped superstructure. The enlarged interlayer distance of MoS_2_ is in favor of alleviating the mechanical stress during the charge and discharge cycles and maintaining the structural integrity. Benefiting from the direct contact between the single-layer MoS_2_ and carbon, the charge transfer rate was obviously enhanced, the resultant C-MoS_2_ sample exhibited faster potassiation and depotassiation kinetics. The MoS_2_ with expended distance and carbon derived from PDDA created more active sites to improve the capacity of C-MoS_2_. As a result, the C-MoS_2_ well addresses the poor cycling stability, low capacity, and low rate performance of MoS_2_ for PIBs. C-MoS_2_-1 exhibited excellent electrochemical properties with higher reversibility specific capacity (437 mAh g^−1^ at 0.1 A g^−1^), superior rate capability (123 mAh g^−1^ at 6.4 A g^−1^), and improved cycle stability (273 mAh g^−1^ after 100 cycles at 1.0 A g^−1^). This work indicates that C-MoS_2_ has great potential as a promising MoS_2_-based anode material for high performance PIBs.

## Figures and Tables

**Figure 1 molecules-28-02608-f001:**
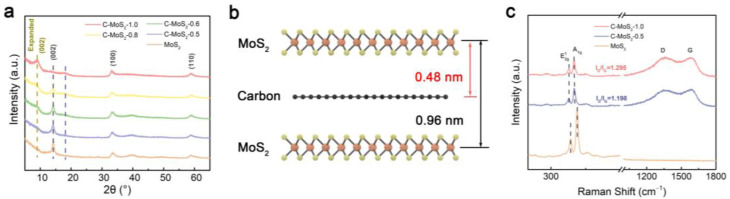
Characterization of samples. (**a**) XRD patterns of the MoS_2_ and C-MoS_2_ samples. (**b**) Crystalline structure of C-MoS_2_-1. (**c**) Raman spectra of the MoS_2_ and C-MoS_2_ samples.

**Figure 2 molecules-28-02608-f002:**
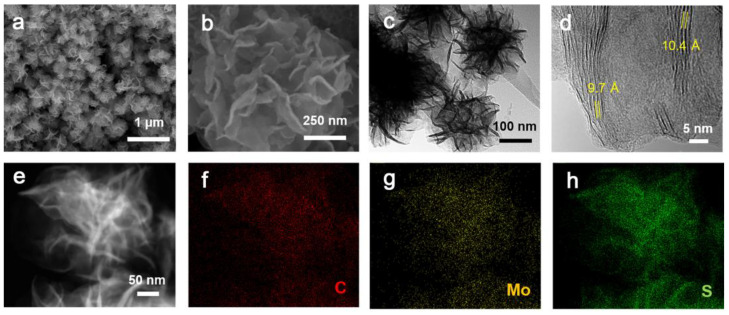
Morphology of samples. (**a**,**b**) SEM images of C-MoS_2_-1. (**c**) Low-resolution TEM of C-MoS_2_-1. (**d**) High-resolution TEM of C-MoS_2_-1. (**e**–**h**) HAADF and EDS of C, Mo, and S for C-MoS_2_-1.

**Figure 3 molecules-28-02608-f003:**
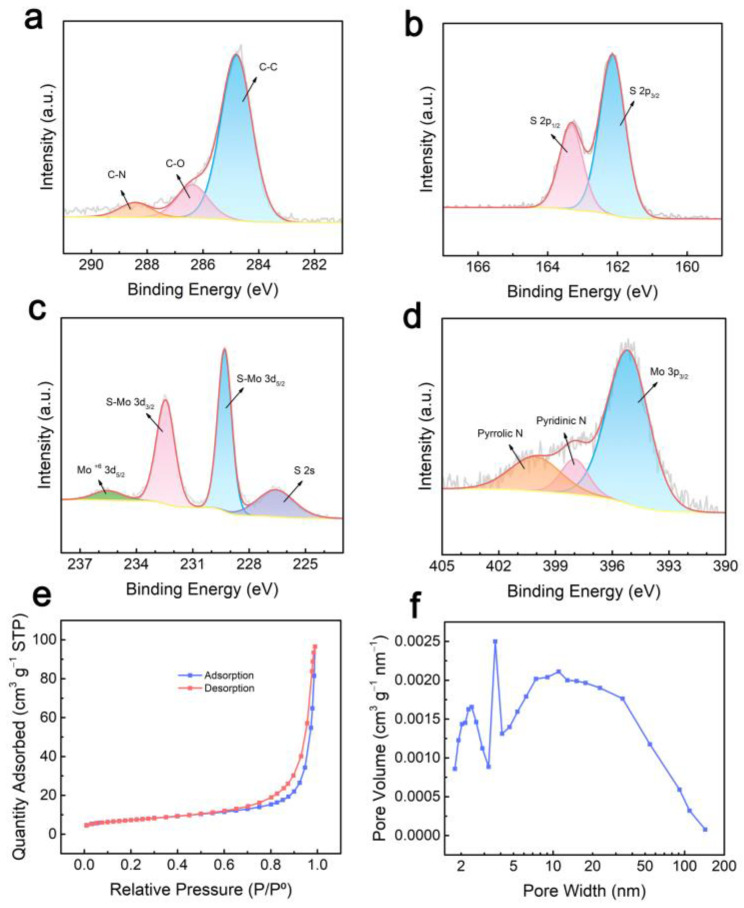
Characterization of samples. (**a**–**d**) High-resolution C 1s, S 2p, Mo 3d, and N 1s XPS spectra of C-MoS_2_-1. (**e**,**f**) N_2_ adsorption–desorption isotherm and BJH pore-size distribution curve of C-MoS_2_-1.

**Figure 4 molecules-28-02608-f004:**
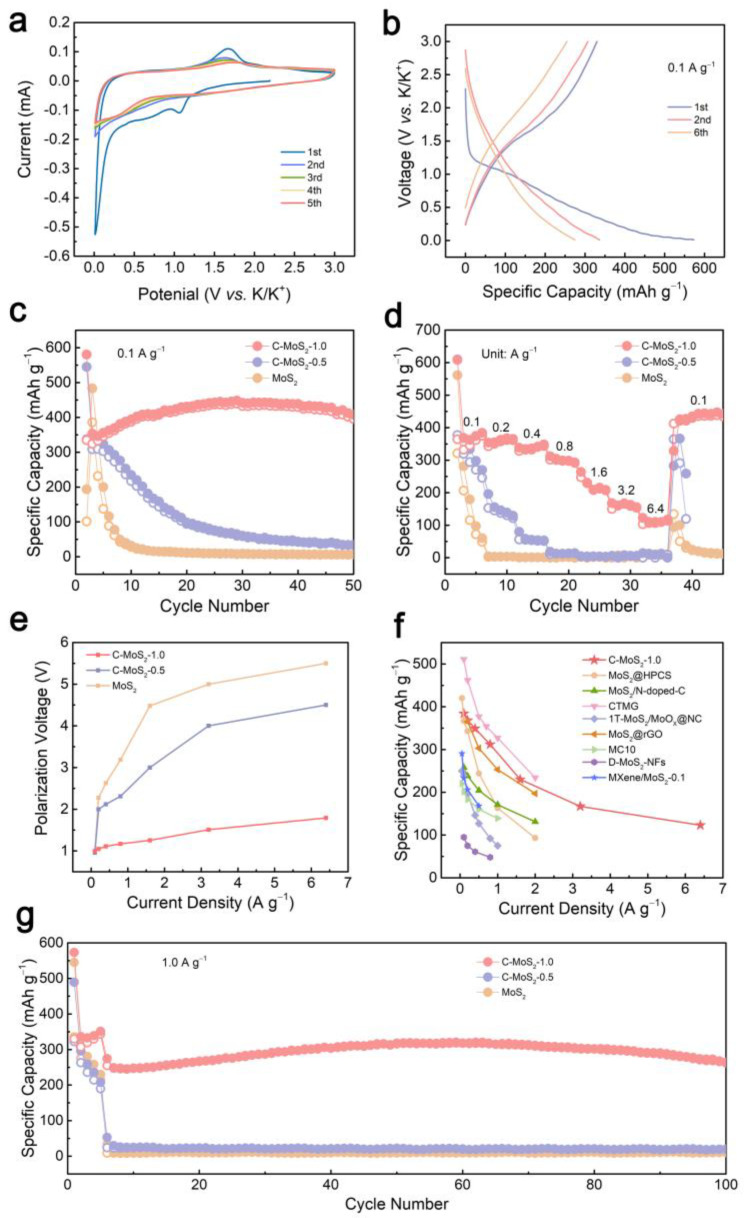
Electrochemical performance of samples. (**a**) CV profiles of C-MoS_2_-1. (**b**) GCD curves of C-MoS_2_-1 electrode at a current density of 0.1 A g^−1^. (**c**) Cycling performance of MoS_2_ and C-MoS_2_ electrodes at 0.1 A g^−1^. (**d**) Rate capability and (**e**) polarization voltage comparison under different current densities of MoS_2_ and C-MoS_2_ electrodes. (**f**) The comparison of the rate capability between C-MoS_2_-1.0 and the other reported MoS_2_-based anodes for PIBs. (**g**) Cycling performance of MoS_2_ and C-MoS_2_ at a current density of 1.0 A g^−1^.

**Figure 5 molecules-28-02608-f005:**
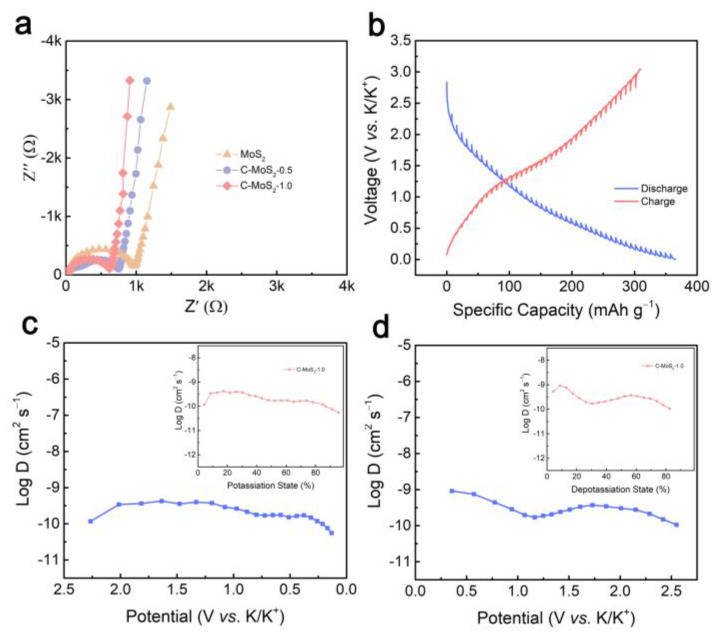
Reaction kinetics of samples. (**a**) Nyquist plots of the MoS_2_ and C-MoS_2_ electrodes. (**b**) GITT test of C-MoS_2_-1 electrode. (**c**,**d**) K^+^ diffusion coefficients from GITT of C-MoS_2_-1 electrode under different voltage. Inset: Diffusion coefficient under different potassiation and depotassiation states.

## Data Availability

Not applicable.

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
