# Peer review of "Interlayer-Expanded MoS2 Enabled by Sandwiched Monolayer Carbon for High Performance Potassium Storage"

_molecules, 2023, doi:10.3390/molecules28062608_

Round 1

Reviewer 1 Report

I found the manuscript "Interlayer-expanded MoS2 enabled by sandwiched monolayer carbon for high performance potassium storage" well written and organized. Therefore, I recommend its publication in Molecules after some minor revisions.

ABSTRACT

- Page 1, line 16: "The interlayer distance of ultrathin C-MoS2 is expanded from 6.2 to 9.6 A due to the formation of MoS2-carbon inter overlapped superstructure". I guess 6.2 A is referred to bare MoS2, but it should be specified.

INTRODUCTION

- Page 1, lines 32-34: "Compared with Na/Na+ (−2.73 V 32 vs. standard hydrogen electrode (SHE)), the K/K+ (−2.93 V vs. SHE) is close to that of Li/Li+ 33 (−3.04 V vs. SHE)". The word "POTENTIAL" is missing.

- Page 2, lines 48-49: Is graphite really good as anode material in potassium-batteries? Its use is reported as controversial by some authors, due to the huge volume expansion. Please, comment on this. 

- Page 1, lines 52-54: "Consideration of the fast potassiation/depotassiation kinetics and relatively high 52 charge/discharge capacities of transition metal dichalcogenides (TMDs), which is becom-53 ing a class of the promising alternatives". The main clause is missing.

RESULTS AND DISCUSSIONS

- Page 5, line 195: CV lines are not "tested" but "recorded".

- Page 5, lines 204 and 205: in the chemical equations, chemical equilibrium arrows should be used instead of resonance arrows. 

MATERIALS AND METHODS: This section should be put before "RESULTS AND DISCUSSIONS". If not possible, please add briefly in the results part how the electrochemical cell is made, for a better comprehension.

Author Response

Response to Reviewer 1 Comments

Reviewer: #1

I found the manuscript "Interlayer-expanded MoS2 enabled by sandwiched monolayer carbon for high performance potassium storage" well written and organized. Therefore, I recommend its publication in Molecules after some minor revisions.

ABSTRACT

1.- Page 1, line 16: "The interlayer distance of ultrathin C-MoS2 is expanded from 6.2 to 9.6 A due to the formation of MoS2-carbon inter overlapped superstructure". I guess 6.2 A is referred to bare MoS2, but it should be specified.

Author’s response: Thanks for your valuable advice. It has been specified in the revised text. Please see the attachment.

INTRODUCTION

2.- Page 1, lines 32-34: "Compared with Na/Na+ (−2.73 V 32 vs. standard hydrogen electrode (SHE)), the K/K+ (−2.93 V vs. SHE) is close to that of Li/Li+ 33 (−3.04 V vs. SHE)". The word "POTENTIAL" is missing.

Author’s response: Sorry for the mistakes. We have corrected it.

3.- Page 2, lines 48-49: Is graphite really good as anode material in potassium-batteries? Its use is reported as controversial by some authors, due to the huge volume expansion. Please, comment on this. 

Author’s response: Graphite is one of the most common commercial LIBs anodes, which has also been investigated as the anode of PIBs in recent years. Kang et al. believe that graphite is one of the most promising anode materials for PIBs [1]. However, the application of graphite anodes in PIBs is limited by the large variation in lattice volume and the low diffusion coefficient of potassium ions during charge/discharge. Various modifications of graphite have been reported, such as doping, expanded layer spacing, morphology modulation, etc. Many modified graphite anodes exhibit satisfactory electrochemical properties for PIBs [2]. So we believe that graphite-based materials are one of the promising anodes for PIBs. “Graphite has been considered one of the most promising anodes” mentioned in the main article is controversial, and has therefore been changed to a “graphite-based anode electrode” in the revised article.

4.- Page 1, lines 52-54: "Consideration of the fast potassiation/depotassiation kinetics and relatively high 52 charge/discharge capacities of transition metal dichalcogenides (TMDs), which is becom-53 ing a class of the promising alternatives". The main clause is missing.

Authors’ response: Sorry for the mistakes. We have corrected it.

RESULTS AND DISCUSSIONS

5.- Page 5, line 195: CV lines are not "tested" but "recorded".

Author’s response: We have corrected it.

6.- Page 5, lines 204 and 205: in the chemical equations, chemical equilibrium arrows should be used instead of resonance arrows. 

Author’s response: We have corrected it.

7.MATERIALS AND METHODS: This section should be put before "RESULTS AND DISCUSSIONS". If not possible, please add briefly in the results part how the electrochemical cell is made, for a better comprehension.

Author’s response: We have put the section of MATERIALS AND METHODS before RESULTS AND DISCUSSIONS, and the meaning of C-MoS2-X is briefly explained in the results part.

References

  1. Lei, Y.; Han, D.; Dong, J.; Qin, L.; Li, X.; Zhai, D.; Li, B.; Wu, Y.; Kang, F. Unveiling the influence of electrode/electrolyte interface on the capacity fading for typical graphite-based potassium-ion batteries. Energy Storage Mater. 2020, 24, 319-328, doi:10.1016/j.ensm.2019.07.043.
  2. Luo, P.; Zheng, C.; He, J.; Tu, X.; Sun, W.; Pan, H.; Zhou, Y.; Rui, X.; Zhang, B.; Huang, K. Structural Engineering in Graphite‐Based Metal‐Ion Batteries. Adv. Funct. Mater. 2021, 32, 2107277, doi:10.1002/adfm.202107277.

Reviewer 2 Report

The manuscript titled “Interlayer-expanded MoS2 enabled by sandwiched monolayer 2 carbon for high performance potassium storage” appears to be technically sound and well-written. Detailed materials characterization and electrochemical analyses have been provided by the authors. However, it would be better to compare the performance with related state-of-the-arts. Moreover, some high-quality articles can be cited in the introduction: e.g., ACS Appl. Mater. Interfaces 2018, 10, 19, 16588–16595; ACS Nano 2014, 8, 5, 4074–4099; Dalton Trans., 2017,46, 15848-15858.

Author Response

Response to Reviewer 2 Comments

Reviewer: #2

The manuscript titled “Interlayer-expanded MoS2 enabled by sandwiched monolayer 2 carbon for high performance potassium storage” appears to be technically sound and well-written. Detailed materials characterization and electrochemical analyses have been provided by the authors. However, it would be better to compare the performance with related state-of-the-arts. Moreover, some high-quality articles can be cited in the introduction: e.g., ACS Appl. Mater. Interfaces 2018, 10, 19, 16588–16595; ACS Nano 2014, 8, 5, 4074–4099; Dalton Trans., 2017,46, 15848-15858.

Author’s response: Very good suggestion. We have added a new figure 4f to compare the rate capability between the C-MoS2-1 and other reported MoS2-based anodes for PIBs. And we have cited these vital works as ref. 24, 26 and 30. Please see the attachment.

Reviewer 3 Report

Pages 5-6

16 – it is obvious that MoS2 is practically not charged individually, so the following questions arise: is it worth studying this anode? Why do the authors not indicate the intercalation reactions of potassium into carbon? It is advisable to compare the results with the work devoted to the K-ion battery with a carbon anode?

17 – what is the reason for the increase in capacity for the C-MoS2-1 sample to 50 cycles?

18 – to lines 242-243 – is such voltage accuracy necessary? What is the measurement error?

Page 8

19 - Carbon was additionally added to the CMoS 2 samples?

20 – in my opinion, the methodological part should definitely be separated from the results and discussions.

On the basis of what data are the synthesis parameters determined? Synthesis time – is it the same for all samples? It is advisable to specify the synthesis reactions or make a link to a detailed description of the technique.

21 – for all reagents used, it is necessary to specify the qualification or purity.

Author Response

Response to Reviewer 3 Comments

Reviewer: #3

Pages 5-6

1.16 – it is obvious that MoS2 is practically not charged individually, so the following questions arise: is it worth studying this anode? Why do the authors not indicate the intercalation reactions of potassium into carbon? It is advisable to compare the results with the work devoted to the K-ion battery with a carbon anode?

Author’s response:

1)“it is obvious that MoS2 is practically not charged individually, so the following questions arise: is it worth studying this anode?”

Although some experimental results show that the bulk MoS2 does not have a high potassium storage capacity, theoretical calculations show that monolayer MoS2 reaches a theoretical capacity of 334.86 mAh g-1 [1], so monolayer or few-layer MoS2 has been reported as more promising anode material for PIBs [2]. In addition, strategies such as expanding the layer spacing of MoS2 and compositing MoS2 with carbon materials have also been demonstrated to be effective methods for improving the electrochemical performance of MoS2-based materials [3,4]. Numerous MoS2-based materials exhibit good electrochemical performance, making these anodes worth studying.

2)“Why do the authors not indicate the intercalation reactions of potassium into carbon? It is advisable to compare the results with the work devoted to the K-ion battery with a carbon anode?”

Very good question, I got your point. In the initial submission, we did not mention the additional part of the capacity provided by carbon, because our samples are mainly MoS2 instead of carbon. In revised manuscript. The sentence “Carbon derived from PDDA create more active sites to improve the capacity of C-MoS2” has been added; Additionally, the comparison of rate capability between the C-MoS2-1 and other reported carbon and MoS2 composites anodes for PIBs has been added as new Figure 4f. Please see the attachment.

2.17 – what is the reason for the increase in capacity for the C-MoS2-1 sample to 50 cycles?

Author’s response: The increased capacity of C-MoS2-1 is mainly due to the increased spacing of the few-layer MoS2 and the addition of carbon providing more reactive sites. To facilitate the reader's comprehension, explanations of the improved capacity have been added to the revised abstract, introduction, results and conclusions parts. 

3.18 – to lines 242-243 – is such voltage accuracy necessary? What is the measurement error?

Author’s response: In the initial submission, we showed the voltage recorded directly from the LAND measurement system, the higher voltage accuracy is not exerting influence on experimental conclusions. However, to keep the consistency of voltage accuracy, only two decimal places are retained in the revised article.

Page 8

4.19 - Carbon was additionally added to the C-MoS-2 samples?

Author’s response: A good point. We have tested the XRD pattern of C-MoS2-1.5 sample, and the XRD of the C-MoS2-1.5 sample is similar to C-MoS2-1 sample. In both of them, the sharp peak located at 14.3° disappears and a dispersive peak at 9.2° appears, which indicates that the distance between (002) crystal plane of MoS2 completely be enlarged from about 6.2 Å to 9.6 Å by the intercalation of carbon. Since we want to modulate the different intercalation states of MoS2 (unintercalated (MoS2), partially intercalation (C-MoS2-0.5) and fully intercalation (C-MoS2-1)) by varying the PDDA content and to investigate their electrochemical performances, we have not tested the electrochemical performance of C-MoS2-2 samples with the same fully intercalated layer as C-MoS2-1.

5.20 – in my opinion, the methodological part should definitely be separated from the results and discussions.

Author’s response: Thanks for your advice. We have simplified the methodology for calculating the diffusion coefficient (D) of K+ in the revised article, and we have retained one equation for readers to understand the calculation of the D.

6. On the basis of what data are the synthesis parameters determined? Synthesis time – is it the same for all samples? It is advisable to specify the synthesis reactions or make a link to a detailed description of the technique.

Author’s response: As soon as the PDDA aqueous solution was added dropwise to the aqueous ammonium molybdate solution, the two were immediately combined due to electrostatic interactions and the solution turned from clear and transparent to a milk-white homogeneous suspension. In order to combine PDDA and molybdate completely, the milk-white homogeneous suspension was stirred for half an hour. All samples differed only in the amount of PDDA added, and the synthesis times were consistent. Since PDDA aqueous solution reacts immediately upon contact with ammonium molybdate, it is very difficult to obtain a linear result for the synthesis time.

7.21 – for all reagents used, it is necessary to specify the qualification or purity.

Author’s response: Thanks for your kind advice. We have added the purity of reagents used in the experimental section.

References

  1. Rehman, J.; Fan, X.; Laref, A.; Dinh, V.A.; Zheng, W.T. Potential anodic applications of 2D MoS2 for K-ion batteries. J. Alloy Compd. 2021, 865, doi:10.1016/j.jallcom.2021.158782.
  2. Wang, H.; Niu, J.; Shi, J.; Lv, W.; Wang, H.; van Aken, P.A.; Zhang, Z.; Chen, R.; Huang, W. Facile Preparation of MoS2 Nanocomposites for Efficient Potassium-Ion Batteries by Grinding-Promoted Intercalation Exfoliation. Small 2021,17, 2102263, doi:10.1002/smll.202102263.
  3. Yao, K.; Xu, Z.; Ma, M.; Li, J.; Lu, F.; Huang, J. Densified Metallic MoS2/Graphene Enabling Fast Potassium‐Ion Storage with Superior Gravimetric and Volumetric Capacities. Adv. Funct. Mater. 2020, 30, 2001484, doi:10.1002/adfm.202001484.
  4. Xie, K.; Yuan, K.; Li, X.; Lu, W.; Shen, C.; Liang, C.; Vajtai, R.; Ajayan, P.; Wei, B. Superior Potassium Ion Storage via Vertical MoS2 "Nano-Rose" with Expanded Interlayers on Graphene. Small 2017, 13, 1701471, doi:10.1002/smll.201701471.

Reviewer 4 Report

For the manuscript titled “Interlayer-expanded MoS2 enabled by sandwiched monolayer

carbon for high performance potassium storage “.  The authors provide an investigation into the performance of  an anode material, MoS2, which has been modified through the expansion of the planar distance.  

The reviewer assessment is that following revisions the manuscript should be accepted for publication.  A few notes for the authors are as follows.

1.      Line 34 - “resulting in the increase in OCV” sounds misleading.  Makes it sound like an increase in OCV over lithiated carbon anodes in Li-ion cells.

2.      Lines 39-41 – can the authors please expand on the safety comment.

3.      Line 42 - the word significant when referring to mass change from Cu to Al.  Possible reference to support this?  Although there is a density difference, I do not think the readers would find this significant.  Perhaps include a mass savings for cm2 to support your claim.

4.      Figure1A – Labels of C-MoS2-1.0, C-MoS2-0.8, C-MoS2-0.6, C-MoS2-0.5 – up until this point in the text the reader is unclear of what these are referring to.  I understand that this is covered in the materials and methods section, but I believe a brief explanation of what these mean would be beneficial prior to showing figures with this nomenclature.

Overall the material is extremely useful, and the manuscript is well prepared.  I find the information is of interest to the readers of Molecules.

Author Response

Response to Reviewer 4 Comments

Reviewer: #4

For the manuscript titled “Interlayer-expanded MoS2 enabled by sandwiched monolayer

carbon for high performance potassium storage “.  The authors provide an investigation into the performance of an anode material, MoS2, which has been modified through the expansion of the planar distance.  

The reviewer assessment is that following revisions the manuscript should be accepted for publication.  A few notes for the authors are as follows.

1.Line 34 - “resulting in the increase in OCV” sounds misleading.  Makes it sound like an increase in OCV over lithiated carbon anodes in Li-ion cells.

Author’s response: You are right. We have corrected it. Please see the attachment.

2.Lines 39-41 – can the authors please expand on the safety comment.

Author’s response: Thanks for your valuable advice. We have expanded on the safety comment.

3.Line 42 - the word significant when referring to mass change from Cu to Al.  Possible reference to support this?  Although there is a density difference, I do not think the readers would find this significant.  Perhaps include a mass savings for cm2to support your claim.

Author’s response: Both Huang and Yamauchi et al. mentioned the weight of the electrode would be reduced when the current collector is replaced from Cu to Al [1,2]. However, the word significant was too absolute, so the controversial word has been removed in the revised article.

4.Figure1A – Labels of C-MoS2-1.0, C-MoS2-0.8, C-MoS2-0.6, C-MoS2-0.5 – up until this point in the text the reader is unclear of what these are referring to.  I understand that this is covered in the materials and methods section, but I believe a brief explanation of what these mean would be beneficial prior to showing figures with this nomenclature.

Author’s response: We have put the section of MATERIALS AND METHODS before RESULTS AND DISCUSSIONS, and the meaning of C-MoS2-X is briefly explained in the results part.

References

  1. Liu, S.; Kang, L.; Henzie, J.; Zhang, J.; Ha, J.; Amin, M.A.; Hossain, M.S.A.; Jun, S.C.; Yamauchi, Y. Recent Advances and Perspectives of Battery-Type Anode Materials for Potassium Ion Storage. ACS Nano 2021, 15, 18931-18973, doi:10.1021/acsnano.1c08428.
  2. Min, X.; Xiao, J.; Fang, M.; Wang, W.; Zhao, Y.; Liu, Y.; Abdelkader, A.M.; Xi, K.; Kumar, R.V.; Huang, Z. Potassium-ion batteries: outlook on present and future technologies. Energy & Environmental Science 2021, 14, 2186-2243, doi:10.1039/d0ee02917c.
